# DP-KAN: Differentially Private Kolmogorov-Arnold Networks

Anonymous Full Paper
Submission 13

## 001 Abstract

We study the Kolmogorov-Arnold Network (KAN), recently proposed as an alternative to the classical Multilayer Perceptron (MLP), in the application for differentially private model training. Using the DP-SGD algorithm, we demonstrate that KAN can be made private in a straightforward manner and evaluated its performance across several datasets. Our results indicate that the accuracy of KAN is not only comparable with MLP but also experiences similar deterioration due to privacy constraints, making it suitable for differentially private model training.

## 1 Introduction

The Kolmogorov-Arnold Network (KAN) [1] has recently emerged as a new approach to symbolic regression and general prediction problems. This new architecture already saw remarkable attention in very recent papers [2–6]. Given its promising capabilities, we find it important to test the ability of KAN to be trained in a privacy-preserving manner without compromising sensitive information. In this work, we chose differential privacy as a way to protect privacy.

As data analysis, and particularly machine learning, continues to expand, the demand for more data increases. However, there is a growing concern about the potential misuse or exploitation of personal information. Differential Privacy [7] is a time-tested notion of formally defined privacy of algorithms concerning sensitive data. Over the past nearly two decades, it has attracted considerable attention from both theoretical and practical perspectives.

The primary result of this work is that **we provide the first integration of KANs with differentially private training algorithms**, specifically with DP-SGD. Our study shows that non-private KANs and MLPs exhibit comparable accuracy. Moreover, when trained with differential privacy, KANs experience similar accuracy degradation compared to MLPs. This makes KANs a promising option for maintaining model performance while ensuring differential privacy in training.

### 1.1 Related Works

Kolmogorov Arnold Networks (KANs) have recently gained significant attention, with various studies building upon the original framework introduced by Ziming et al. [1]. Zavareh and Chen [6] introduced Wav-KAN, enhancing interpretability and performance by integrating wavelet functions to efficiently capture both high-frequency and low-frequency components of input data. For time series forecasting, Genet and Inzirillo [2] proposed Temporal KAN (TKAN), combining the strengths of Long Short-Term Memory (LSTM) networks and KANs, while Vaca-Rubio et al. [4] demonstrated that KANs outperform conventional MLPs in satellite traffic forecasting with fewer parameters. Xu et al. [5] introduced FourierKAN-GCF, a graph-based recommendation model that utilizes a Fourier KAN to improve feature transformation during message passing in graph convolution networks (GCNs), achieving superior performance in recommendation tasks. A benchmarking study by Poeta et al. [8] on real-world tabular datasets shows that KANs achieve superior or comparable accuracy and F1 scores compared to MLPs, especially on larger datasets, though with higher computational costs. Other notable advancements include Shukla et al. [3], which compared KANs with traditional Multi-Layer Perceptrons (MLPs), and Basim and Naveed [9], which developed Convolutional KAN (ConvKAN) for enhanced image processing.

Differentially private stochastic gradient descent (DP-SGD) has become a standard in private machine learning, being applied in both convex and non-convex optimization. Bassily et al. [10] demonstrated its effectiveness in convex optimization, while Abadi et al. [11] extended its application to deep learning, ensuring privacy in training deep neural networks. In our study, we use DP-Adam, a specific member of the DP-SGD family of algorithms, to leverage its adaptive learning rate capabilities for improved performance. The DP-Adam method has been studied in various works, including Gylberth et al. [12], who demonstrated improved accuracy and faster convergence, and Tang et al. [13], who proposed DP-AdamBC to correct bias in the second moment estimation. In differentially private regression, our tabular data experiments build on the work by Amin et al. [14], who introduced a method using the exponential mechanism to select a model with high Tukey depth, eliminating the need for data bounds or hyperparameter selection. Other significant contribution includes Alabi et al. [15], who designed differentially private algorithms for simple linear regression in small datasets.

## 2 Background

### 2.1 Kolmogorov-Arnold Networks

Kolmogorov-Arnold Networks (KAN) [1] are a novel neural network architecture inspired by the Kolmogorov-Arnold representation theorem [16], which states that any multivariate continuous function $f : [0,1]^n \to \mathbb{R}$ can be decomposed into a sum of compositions of univariate functions:

$$f(x^{(1)}, \ldots, x^{(n)}) = \sum_{q=1}^{2n+1} \Phi_q \left( \sum_{p=1}^{n} \phi_{q,p}(x^{(p)}) \right). \quad (1)$$

KANs replace fixed activation functions with learnable univariate functions parameterized as B-splines. This enhances flexibility and interpretability compared to traditional Multi-Layer Perceptrons (MLPs), where activations occur at the nodes and weights are linear.

In KANs, B-splines serve as the building blocks for learnable univariate functions $\phi_{q,p}$, expressed as:

$$\mathrm{spline}(x) = \sum_i c_i B_i(x), \quad (2)$$

where $c_i$ are coefficients learned during training and $B_i(x)$ are B-spline basis functions. KANs use residual activation functions, combining a basis function $b(x)$ with the B-spline:

$$\phi(x) = w_b b(x) + w_s \mathrm{spline}(x), \quad (3)$$

where $b(x)$ is a sigmoid linear unit (SiLU), and $w_b$ and $w_s$ are trainable weights.

KANs are trained using backpropagation, compatible with standard optimization techniques such as Differentially Private SGD (DP-SGD 1). They have shown superior performance to MLPs in specific tasks, such as formula recovery and symbolic regression [1].

### 2.2 Differential Privacy

Differential privacy guarantees that a (randomized) algorithm is *stable* with respect to single data point changes of the input dataset. In particular, we say that two datasets $D$ and $D'$ are adjacent if they differ only in one sample. Then, we have the following

**Definition 2.1 ($(\varepsilon, \delta)$-differential privacy [7])** *A randomized algorithm $\mathcal{A} : \mathcal{D} \to \mathcal{S}$ satisfies $(\varepsilon, \delta)$-differential privacy if for any two adjacent inputs $D, D' \in \mathcal{D}$, and for any subset of the output space $S \subseteq \mathcal{S}$, we have*

$$\mathbb{P}\left( \mathcal{A}(D) \in S \right) \leq e^\varepsilon \mathbb{P}\left( \mathcal{A}(D') \in S \right) + \delta. \quad (4)$$

Here, $\mathcal{D}$ represents the space of all datasets, and $\mathcal{S}$, in the case of deep learning models, is the space of the

---

**Algorithm 1** Differentially private gradient descent with Adam Optimizer

---

**Input:** Number of iterations $T$, learning rate $\eta$, clipping constant $C$, *noise multiplier* $\sigma$, batch size $B$, initialization $\theta_0$.

1: **for** $t \in [T]$ **do**
2:     Randomly sample a batch of $B$ samples
3:     Compute the gradients
        $g(x_i, y_i, \theta_{t-1}) \leftarrow \nabla_\theta \ell(x_i, y_i, \theta_{t-1})$
4:     Clip the gradients
        $\tilde{g}(x_i, y_i, \theta_{t-1}) \leftarrow g(x_i, y_i, \theta_{t-1})/\zeta,$
        where $\zeta = \max\left(1, \frac{\|g(x_i, y_i, \theta_{t-1})\|_2}{C}\right)$
5:     Aggregate the noisy gradients
        $g_t \leftarrow \frac{1}{B} \sum_i \tilde{g}(x_i, y_i, \theta_{t-1}) + \frac{\sigma C}{B} \mathcal{N}(0, I)$
6:     Update the model parameters
        $\theta_t = \mathrm{Adam}(\theta_{t-1}, g_t)$
7: **end for**

**Output:** Model parameters $\theta_T$

---

trainable parameters. Differential privacy ensures that from the resulting model, one cannot extract the training data with high probability, nor can one determine with confidence which specific data points were used during training. This property is crucial for protecting individual privacy in machine learning applications, particularly in contexts where sensitive data may be involved.

In machine learning, a well-established family of algorithms allows us to train models with differential privacy (DP) guarantees: differentially private stochastic gradient descent (DP-SGD). These algorithms [11] involve an iterative process where gradients are clipped to have a bounded maximum norm and then summed up with Gaussian noise scaled by a *noise multiplier* to ensure given privacy guarantees. The algorithm described in 1 builds on the Adam optimizer, illustrating one specific example within this family.

## 3 Results

We conducted two sets of experiments: regression on tabular data and classification on the MNIST and USPS image datasets. For the regression task, we used various tabular datasets [1] from Amin et al. [14] and trained differentially private and non-private models. Specifically, we used mean squared error and a one-layer neural network model for linear regression and one layer KAN in our experiments. We used the datasets and training settings from Amin et al. [14]. On each of the datasets and each of the models, we computed the coefficient of determination ($R^2$ score). The results can be found in Table 1 and the hyperparameters in Table A.3. KAN

---

[1] We were unable to reproduce the results for the Beijing dataset, so it has not been included in our analysis.

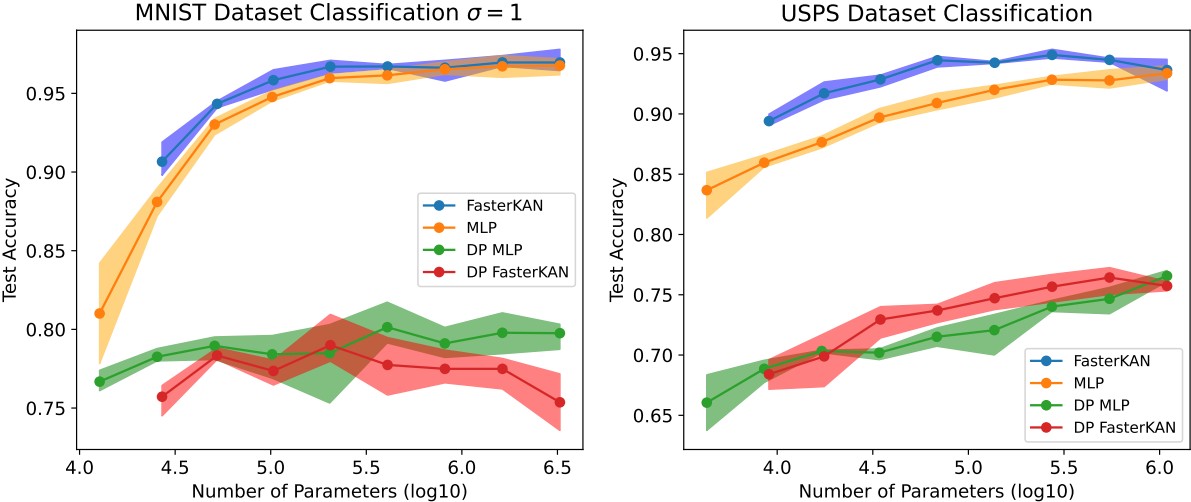

**Figure 1.** Validation accuracy vs the number of parameters for FasterKAN, MLP, DP MLP, and DP FasterKAN on MNIST and USPS datasets. The privacy garantees for the MNIST models are $(0.87, 10^{-5})$ DP and $(2.03, 10^{-5})$ DP for USPS dataset. The error bars are based on three trials for each point.

model demonstrated lower quality degradation due to privacy across most datasets.

We also trained differentially private and non-private models on the MNIST and USPS datasets using the fasterKAN [17]. This implementation has a superior computational speed compared to the official pykan implementation [1], which was found to be inefficient and impractical for large datasets. For differential privacy, we employed the DP Adam optimizer via the pyvacy library [18], an unofficial PyTorch adaptation of TensorFlow Privacy, which was more PyTorch compatible with fasterKAN [17]. We compared the quality of fasterKAN against a Multi-Layer Perceptron (MLP) in a setting of two layers networks with varying hidden layer sizes, resulting in different numbers of parameters. We used CrossEntropyLoss as the loss function and the accuracy on the test dataset for evaluation, the hyperparameters of those experiments can be found in the Table A.4. The results, shown in Figure A.1, indicate that fasterKAN consistently achieved higher accuracy for a relatively lower number of parameters compared to the MLP models. In the differentially private setting, fasterKAN suffered a similar quality degradation to the MLP models. This demonstrates the effectiveness of fasterKAN in balancing accuracy and privacy constraints, making it an appropriate choice for differentially private training scenarios.

## 4 Conclusion

In this study, we showed that the Kolmogorov-Arnold Network (KAN) is a possible alternative to the classical Multi-Layer Perceptron (MLP) for differentially private training scenarios. Through experiments on both tabular data and MNIST and USPS image classification tasks, KAN not only achieves comparable accuracy with MLP but also shows similar performance under privacy constraints. Future work can explore further optimizations and applications of KAN in various privacy-preserving machine learning contexts.

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

| Dataset | Linear Reg | KAN | DP-SGD Reg | DP-KAN | Lin Reg drop, % | KAN drop, % |
|---------|-----------|-----|-----------|--------|----------------|-------------|
| Synthetic | 0.997 | 0.996 | $0.997 \pm 0.001$ | $0.953 \pm 0.035$ | **0.1** | 4.3 |
| California | 0.637 | 0.652 | $0.091 \pm 0.009$ | $0.172 \pm 0.020$ | 85.7 | **73.7** |
| Diamonds | 0.907 | 0.926 | $0.829 \pm 0.000$ | $0.924 \pm 0.000$ | 8.6 | **0.3** |
| Traffic | 0.966 | 0.979 | $0.937 \pm 0.002$ | $0.940 \pm 0.000$ | **3.0** | 4.0 |
| NBA | 0.622 | 0.631 | $0.529 \pm 0.051$ | $0.572 \pm 0.015$ | 15.0 | **9.26** |
| Garbage | 0.546 | 0.554 | $0.275 \pm 0.027$ | $0.371 \pm 0.148$ | 49.6 | **33.0** |
| MLB | 0.722 | 0.723 | $0.714 \pm 0.004$ | $0.719 \pm 0.002$ | 1.1 | **0.6** |

**Table 1.** $R^2$ scores for different models in the experimental setup of Amin et al. [14]. For each of the datasets we computed the relative drop in quality due to privacy. We provide error intervals based on three trials, calculated as half the difference between the maximum and minimum values. We also observe that there is almost no stochasticity in non-private models; therefore, we do not include errors for them.

preprint arXiv:2406.01034. 2024. DOI: 10.48550/arXiv.2406.01034.

[6] B. Zavareh and H. Chen. *Wav-KAN: Wavelet Kolmogorov-Arnold Networks*. arXiv preprint arXiv:2405.12832. 2024. DOI: 10.48550/arXiv.2405.12832.

[7] C. Dwork. "Differential privacy". In: *International Colloquium on Automata, Languages, and Programming (ICALP)*. 2006. DOI: 10.1007/11787006_1.

[8] E. Poeta, F. Giobergia, E. Pastor, T. Cerquitelli, and E. Baralis. *A Benchmarking Study of Kolmogorov-Arnold Networks on Tabular Data*. arXiv preprint arXiv:2406.14529. 2024. DOI: 10.48550/arXiv.2406.14529.

[9] A. Basim and A. Naveed. *Suitability of KANs for Computer Vision: A preliminary investigation*. arXiv preprint arXiv:2406.09087. 2024. DOI: 10.48550/arXiv.2406.09087.

[10] R. Bassily, A. Smith, and A. Thakurta. "Private empirical risk minimization: Efficient algorithms and tight error bounds". In: *IEEE Symposium on Foundations of Computer Science (FOCS)*. 2014. DOI: 10.1109/FOCS.2014.56.

[11] M. Abadi, A. Chu, I. Goodfellow, H. B. McMahan, I. Mironov, K. Talwar, and L. Zhang. "Deep learning with differential privacy". In: *ACM Conference on Computer and Communications Security (CCS)*. 2016. DOI: 10.1145/2976749.2978318.

[12] R. Gylberth, R. Adnan, S. Yazid, and T. Basaruddin. "Differentially private optimization algorithms for deep neural networks". In: *Advanced Computer Science and Information Systems (ICACSIS)* (2017). DOI: 10.1109/ICACSIS.2017.8355063.

[13] Q. Tang, F. Shpilevskiy, and M. Lécuyer. "Your DP-Adam Is Actually DP-SGD (Unless You Apply Bias Correction)". In: *Conference on Artificial Intelligence (AAAI)* (2024). DOI: 10.1609/aaai.v38i14.29451.

[14] K. Amin, M. Joseph, M. Ribero, and S. Vassilvitskii. "Easy Differentially Private Linear Regression". In: *International Conference on Learning Representations (ICLR)*. 2022. DOI: 10.2478/popets-2022-0041.

[15] D. Alabi, A. McMillan, J. Sarathy, A. Smith, and S. Vadhan. "Differentially Private Simple Linear Regression." In: *Privacy Enhancing Technologies Symposium (PETS)*. 2022. DOI: 10.2478/popets-2022-0041.

[16] A. Kolmogorov. "On the representation of continuous functions of several variables by superpositions of continuous functions of a smaller number of variables". In: *American Mathematical Society* (1961). DOI: 10.1007/978-3-642-01742-1_5.

[17] A. Delis. *FasterKAN*. https://github.com/AthanasiosDelis/faster-kan/. 2024.

[18] C. Waites. *pyvacy*. https://github.com/ChrisWaites/pyvacy. 2019.

# A Appendix

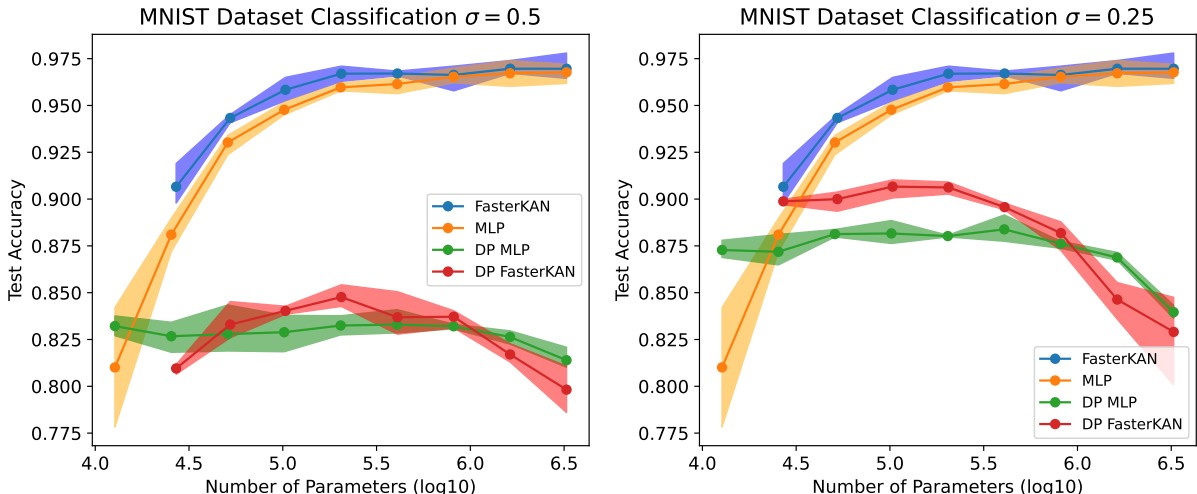

**Figure A.1.** Validation accuracy versus the number of parameters on MNIST for FasterKAN, MLP, DP MLP, and DP FasterKAN at different noise levels $\sigma \in \{0.5, 0.25\}$, which corresponds to $(\{7.5, 121\}, 10^{-5})$ Differential Privacy.

| Model | Width | Parameters | Test Accuracy | DP Test Accuracy | | |
|-------|-------|-----------|---------------|------------------|---|---|
| | | | | $\sigma = 1$ | $\sigma = 0.5$ | $\sigma = 0.25$ |
| KAN | 2048 | $3,257,888$ | $0.970 \pm 0.008$ | $0.754 \pm 0.018$ | $0.814 \pm 0.005$ | $0.840 \pm 0.003$ |
| | 1024 | $1,629,728$ | $0.970 \pm 0.004$ | $0.775 \pm 0.007$ | $0.826 \pm 0.004$ | $0.869 \pm 0.002$ |
| | 512 | $815,648$ | $0.966 \pm 0.005$ | $0.775 \pm 0.012$ | $0.832 \pm 0.001$ | $0.876 \pm 0.002$ |
| | 256 | $408,608$ | $0.967 \pm 0.001$ | $0.775 \pm 0.017$ | $0.833 \pm 0.006$ | $0.884 \pm 0.007$ |
| | 128 | $205,088$ | $0.967 \pm 0.004$ | $0.777 \pm 0.020$ | $0.832 \pm 0.005$ | $0.880 \pm 0.000$ |
| | 64 | $103,328$ | $0.958 \pm 0.007$ | $0.774 \pm 0.007$ | $0.829 \pm 0.010$ | $0.882 \pm 0.006$ |
| | 32 | $52,448$ | $0.943 \pm 0.002$ | $0.783 \pm 0.005$ | $0.828 \pm 0.012$ | $0.881 \pm 0.002$ |
| | 16 | $27,008$ | $0.907 \pm 0.012$ | $0.757 \pm 0.007$ | $0.827 \pm 0.008$ | $0.872 \pm 0.008$ |
| MLP | 4096 | $3,256,330$ | $0.968 \pm 0.004$ | $0.798 \pm 0.006$ | $0.832 \pm 0.005$ | $0.873 \pm 0.005$ |
| | 2048 | $1,628,170$ | $0.967 \pm 0.007$ | $0.798 \pm 0.013$ | $0.798 \pm 0.012$ | $0.829 \pm 0.023$ |
| | 1024 | $814,090$ | $0.965 \pm 0.005$ | $0.791 \pm 0.010$ | $0.817 \pm 0.005$ | $0.846 \pm 0.010$ |
| | 512 | $407,050$ | $0.961 \pm 0.004$ | $0.801 \pm 0.016$ | $0.837 \pm 0.005$ | $0.882 \pm 0.008$ |
| | 256 | $203,530$ | $0.960 \pm 0.002$ | $0.785 \pm 0.018$ | $0.837 \pm 0.011$ | $0.896 \pm 0.002$ |
| | 128 | $101,770$ | $0.948 \pm 0.004$ | $0.784 \pm 0.012$ | $0.848 \pm 0.006$ | $0.906 \pm 0.003$ |
| | 64 | $50,890$ | $0.930 \pm 0.004$ | $0.790 \pm 0.006$ | $0.840 \pm 0.002$ | $0.907 \pm 0.005$ |
| | 32 | $25,450$ | $0.881 \pm 0.009$ | $0.783 \pm 0.005$ | $0.833 \pm 0.010$ | $0.900 \pm 0.005$ |
| | 16 | $12,730$ | $0.810 \pm 0.032$ | $0.767 \pm 0.007$ | $0.810 \pm 0.003$ | $0.899 \pm 0.001$ |

**Table A.1.** Test Accuracy for FasterKAN, MLP, DP MLP, and DP FasterKAN on **MNIST** based on 3 trials.

| Model | Width | Parameters | Test Accuracy | DP Test Accuracy |
|---|---|---|---|---|
| | 16 | $4,282$ | $0.837 \pm 0.019$ | $0.661 \pm 0.023$ |
| | 32 | $8,554$ | $0.860 \pm 0.005$ | $0.689 \pm 0.009$ |
| | 64 | $17,098$ | $0.877 \pm 0.005$ | $0.703 \pm 0.001$ |
| | 128 | $34,186$ | $0.897 \pm 0.006$ | $0.702 \pm 0.004$ |
| MLP | 256 | $68,362$ | $0.909 \pm 0.007$ | $0.715 \pm 0.008$ |
| | 512 | $136,714$ | $0.920 \pm 0.005$ | $0.721 \pm 0.017$ |
| | 1024 | $273,418$ | $0.928 \pm 0.003$ | $0.740 \pm 0.005$ |
| | 2048 | $546,826$ | $0.928 \pm 0.008$ | $0.747 \pm 0.011$ |
| | 4096 | $1,093,642$ | $0.934 \pm 0.006$ | $0.766 \pm 0.005$ |
| | 16 | $9,056$ | $0.894 \pm 0.005$ | $0.684 \pm 0.012$ |
| | 32 | $17,600$ | $0.917 \pm 0.007$ | $0.699 \pm 0.022$ |
| | 64 | $34,688$ | $0.929 \pm 0.005$ | $0.729 \pm 0.013$ |
| | 128 | $68,864$ | $0.944 \pm 0.004$ | $0.737 \pm 0.008$ |
| KAN | 256 | $137,216$ | $0.942 \pm 0.001$ | $0.747 \pm 0.011$ |
| | 512 | $273,920$ | $0.949 \pm 0.004$ | $0.757 \pm 0.011$ |
| | 1024 | $547,328$ | $0.945 \pm 0.002$ | $0.764 \pm 0.011$ |
| | 2048 | $1,094,144$ | $0.936 \pm 0.013$ | $0.757 \pm 0.004$ |

**Table A.2.** Test Accuracy for FasterKAN, MLP, DP MLP, and DP FasterKAN on **USPS** dataset based on 3 trials.

| Dataset | Noise mult. | DP KAN | | | | DP Linear Reg | | | |
|---|---|---|---|---|---|---|---|---|---|
| | | Epochs | C. N. | Lr | Bs | Epochs | C. N. | Lr | Bs |
| Synthetic | 1.472 | 20 | 1 | 1 | 128 | 20 | 1 | 1 | 128 |
| California | 1.178 | 20 | 100 | 1 | 64 | 20 | 100 | 1 | 64 |
| Diamonds | 1.089 | 20 | $10^6$ | 1 | 128 | 20 | $10^6$ | 1 | 128 |
| Traffic | 2.016 | 1 | $10^6$ | 1 | 1024 | 1 | $10^6$ | 1 | 1024 |
| NBA | 2.468 | 20 | 100 | 1 | 512 | 20 | 100 | 1 | 512 |
| Garbage | 0.998 | 20 | 1 | 1 | 32 | 20 | 1 | 1 | 32 |
| MLB | 1.066 | 10 | 100 | 0.01 | 512 | 10 | 100 | 0.01 | 512 |

**Table A.3.** The table presents the hyperparameters for tabular data experiments, including the Noise multiplier, number of epochs, the Clipping Norm (C. N.), the learning rate (Lr), and the batch size (Bs), for both Adam and DP Adam optimizers. For KAN, we use a grid size of 2 and a value of k equal to 2. We ensure $(\log(3), 10^{-5})$-differential privacy in the same manner as described in [14].

| Method | Epochs | Batch Size | Noise Multiplier | Clipping Norm | Learning Rate |
|---|---|---|---|---|---|
| FasterKAN | 15 | 64 | 0 | - | 0.001 |
| MLP | 15 | 64 | 0 | - | 0.001 |
| DP FasterKAN | 15 | 64 | $\{1, 0.5, 0.25\}$ | $10^{-3}$ | 0.001 |
| DP MLP | 15 | 64 | $\{1, 0.5, 0.25\}$ | $10^{-3}$ | 0.001 |

**Table A.4.** Hyperparameters for MNIST and USPS experiments. Number of epochs, batch size, noise multiplier, max grad norm, learning rate. Trained with AdamW and DP Adam. FasterKAN with parameters grid_min =-1.2, grid_max = 0.2, num_grids = 2, exponent = 2, inv_denominator = 0.5, train_grid =False, train_inv_denominator=False. We used batch clipping for non-private training to ensure the stability of the optimization.

