# OpenReview forum: "DP-KAN: Differentially Private Kolmogorov-Arnold Networks"
_NLDL.org/2025/Conference — Submitted to NLDL 2025_

### Official Review · Reviewer_SnD9 · 2024-09-22
**Review for DP-KAN: Differentially Private Kolmogorov-Arnold Networks**

**Confidence:** 5

**Summary:**

This study explores the Kolmogorov-Arnold Network (KAN) as an alternative to the classical Multilayer Perceptron (MLP) for differentially private model training. Using the DP-SGD algorithm, KAN was made differentially private and evaluated across several datasets. The results show that KAN achieves accuracy comparable to MLP and experiences similar performance deterioration under privacy constraints, making it suitable for privacy-preserving machine learning.

**Strengths:**

- Paper is clearly written
- Contribution is clearly mentioned. "first integration of KANs with differentially private training algorithms"

**Weaknesses:**

- Contribution is weak, It is not clear what was the challenge in addressing the research question?
- Experiments could have been more elaborate in the setting,
    - From DP side - probably adding more budget, more DP variations - these could have made the paper more concrete in the claims.
    - From KAN side - May be more variations in experiments as mentioned in the related works.
    - Implementation details are missing. If the implementation produced some challenges, that could have been noted down as well.

**Final Rebuttal Confidence:**

5

**Final Rebuttal Justification:**

Based on the rebuttal by authors and discussion among the reviewers and AC.

**Justification:**

Even though contribution is weak in terms of just addition of the DPSGD (Adam) to KANs, paper could have been more complete from the perspective of experiments. Currently that is lacking. Hence the evaluation of the paper.

With the addition of more comprehensive set of experiments, the paper can be made better.

---

> ### Author Rebuttal · Authors · 2024-10-21
>
> We thank the reviewer for their valuable feedback.
>
> **\> Regarding the challenge in addressing the research question:**
>
> We thank the reviewer for this important question. While it is indeed true that our work builds on two existing frameworks (differentially private training and Kolmogorov-Arnold networks), the challenge of this research question lies in its effective implementability. Unlike standard training approaches, where the methodology is well-established for both MLPs and non-private training, private training is not as mature and presents greater challenges. Implementing DP-SGD in deep learning requires carefully balancing hyperparameters such as the clipping constant, the noise multiplier, and the number of training iterations, where suboptimal choices can often hinder successful optimization. Furthermore, private training is computationally more intensive in terms of both computing time and memory usage, making fine hyperparameter optimizations even infeasible for larger models and datasets (see [1]).
>
>
> **\>Regarding the limitation of our experimental results:**
>
> We thank the reviewer for pointing this out. We received similar feedback also from Reviewers D8gj and VhGM, and we are glad to provide additional experiments: we include in the revised version experiments on the USPS dataset, as well as MNIST with more diverse DP parameter settings.
>
> Please refer to the revised manuscript to see the updates.
>
> [1] https://arxiv.org/abs/2204.13650

---

### Official Review · Reviewer_VhGM · 2024-10-07
**Experimental results on DP-SGD in KAN are interesting, but questions remain**

**Confidence:** 3

**Summary:**

The paper studies the use of differentially private SGD on Kolmogorov-Arnold networks and compare results to MLPs and linear regression. The results are, while not surprising, of practical interest but questions remain about the test and the authors interpretation of the results.

**Strengths:**

* The paper adress an interesting problem, namely the performance of Kolmogorov Arnold networks using differentially private gradient descent.
* The background and approach are briefly but concisely presented.
* The results provide insight into the performance of KAN using DP-SGD

**Weaknesses:**

* The results section, the most interesting part of the paper, is very brief. More details on experiments and datasets used are necessary.
* The paper would benefit from more analysis of the results.
* While interesting, the tests comparing KAN to MLP are not very comprehensive.
* There are questions around the interpretation of the results: Mayby I am missing something, but how is the statement "faster KAN did not suffer as much quality degradation as the MLP models" referring to figure 1 true?

**Justification:**

While potentially providing practical experimental insight on the performance of KAN using DP-SGD, the evaluations are a bit too limited and too briefly described to be truly useful. Further, some interpretation of the results potentially put other interpretations and methodology into question.

---

> ### Author Rebuttal · Authors · 2024-10-21
>
> We thank the reviewer for their valuable feedback.
>
> Regarding the first three pointed out weaknesses, we are glad to provide additional experiments, including the USPS dataset, as well as MNIST with more diverse privacy parameter settings, and to discuss the results in greater detail. See Figure 1 and A.1 in the revised version.
>
> Regarding the phrase “did not suffer as much”, we thank the reviewer for pointing this out (as for Reviewer D8gj). It indeed contradicts the results and conclusions we presented in paper (see abstract, introduction, and conclusion sections). We corrected this typo in the revised version.
>
> Please refer to the revised manuscript to see the updates.

---

### Official Review · Reviewer_D8gj · 2024-10-09
**An empirical comparison between privacy-preserving training of MLPs and KANs**

**Confidence:** 4

**Summary:**

This paper studies training of Kolmogorov-Arnold Networks (KANs) under the privacy protection by differential privacy (DP). DP is known to degrade the utility of the learning in order to protect the privacy of the individuals. This paper studies empirically whether this utility hit in deep learning can be reduced by using another function approximator than MLP. Authors apply DP stochastic gradient descent (DP-SGD) for KANs, and empirically compare the obtained accuracy against DP-SGD trained MLPs in regression and classification tasks with multiple data sets. The results suggest that KANs trained with DP-SGD suffer a smaller utility loss, caused by DP, than the MLPs.

**Strengths:**

The empirical results demonstrate that for certain machine learning tasks, training a KAN instead of an MLP can help improve the privacy utility trade-off. This is an interesting, and as far as I'm aware, a novel contribution.

The main method is presented clearly in the paper, and as authors use readily available packages for DP deep learning, I also believe the experimental results are correct.

**Weaknesses:**

The discussion of the results could be improved. Since DP-SGD introduces stochasticity into the learning outcomes, it would be important to repeat results of multiple repeats of the method in Table 1. This would help to assess if the lower utility degradation of the DP-KAN is actually statistically significant, or is it just a fluke. For the MNIST experiment you seem to report some error bars for the plots, however you do not explicitly state what these bars are.

Regarding the MNIST experiment. You write that "In the differentially private setting, fasterKAN did not suffer as much quality degradation as the MLP models". Is there an error in the labels? To me, it seems that the DP MLP is almost always always slightly above the DP FasterKAN, while the FasterKAN dominates the non-DP results. So how can it be then that DP MLP has a larger degradation in utility?

Although I believe the implementation is correct, it would be great if authors could clearly state the KAN loss function in the paper. As DP-SGD relies on loss function to be composable into a sum over the individuals, it would help the clarity if authors could explicitly say that the target loss is composable over the per-example losses in Eq. (1). This could be fixed by e.g. using different indexing than $x_i$ to denote the $i$th dimension (in Eq. (1)) and $i$th sample (in Alg. 1).

**Justification:**

While the studying the different trade-offs between private training of MLPs vs. KANs is an interesting research direction, I believe the paper should provide more rigorous evidence on its claims. The most crucial thing would be to test multiple repeats of the DP-SGD algorithm to test if the differences between DP KAN and DP MLP are actually statistically significant.

---

> ### Author Rebuttal · Authors · 2024-10-21
>
> We thank the reviewer for their valuable feedback.
>
> **\>Response to the Weakness in the error bars:**
>
> We agree with the Reviewer that the current presentation of Table 1 lacks information on the uncertainty of the shown quantities. We are happy to provide a more detailed discussion of the results, as well as to include error bars in the first table.
> We finally remark that in Table A1, we have indicated that the error bars are based on three independent trials for each data point; however, we are happy to include this information in the main body as well.
>
> **\>Response to the Weakness in the Phrase "Did Not Suffer as Much"**
>
> We thank the reviewer for pointing out this typo (as for Reviewer VhGM). This sentence is in clear contradiction with both the results and conclusions we presented during the paper (see abstract, introduction, and conclusion sections). We edited this statement in the revised version.
>
> **\>Response to the Weakness in the KAN loss**
>
> We thank the reviewer for this reasonable suggestion to improve readability. We remark that we train using a standard cross-entropy loss for image classification, and we revised the indexing in Equation (1) to avoid confusion between different samples and dimensions.
>
> Please refer to the revised manuscript to see the updates.

---

### Meta-Review · Area_Chair_ibJ4 · 2024-10-30

**Recommendation:** Reject
**Confidence:** 4

**Metareview:**

The authors present privacy-aware Kolmogorov-Arnold Networks (KANs), i.e. KANs trained with differential privacy, ensuring privacy without compromising accuracy. KANs exhibit similar accuracy degradation to MLPs when trained with differential privacy, making them a promising option for privacy-preserving model training. Differentially private stochastic gradient descent (DP-SGD), used in this work, is a well-established family of algorithms that allows training models with differential privacy guarantees.

However, reviewers unanimously highlighted that the paper could have been better written, e.g. the appendix could have been part of the main paper, implementation details should have been clearer, some more experiments could have been provided, etc.

This line of work has definitely merits, and hopefully, the authors can submit an improved version of this paper to another venue in the coming period.

**Suggested Changes To The Recommendation:**

2: I'm certain of the recommendation.  It should not be changed

---

### Decision · Program_Chairs · 2024-11-06

Reject